# Graph Transformer Networks

**Seongjun Yun, Minbyul Jeong, Raehyun Kim, Jaewoo Kang* , Hyunwoo J. Kim***
Department of Computer Science and Engineering
Korea University
{ysj5419, minbyuljeong, raehyun, kangj, hyunwoojkim}@korea.ac.kr

## Abstract

Graph neural networks (GNNs) have been widely used in representation learning on
graphs and achieved state-of-the-art performance in tasks such as node classification
and link prediction. However, most existing GNNs are designed to learn node
representations on the *fixed* and *homogeneous* graphs. The limitations especially
become problematic when learning representations on a misspecified graph or
a *heterogeneous* graph that consists of various types of nodes and edges. In
this paper, we propose Graph Transformer Networks (GTNs) that are capable of
generating new graph structures, which involve identifying useful connections
between unconnected nodes on the original graph, while learning effective node
representation on the new graphs in an end-to-end fashion. Graph Transformer layer,
a core layer of GTNs, learns a soft selection of edge types and composite relations
for generating useful multi-hop connections so-called *meta-paths*. Our experiments
show that GTNs learn new graph structures, based on data and tasks without
domain knowledge, and yield powerful node representation via convolution on the
new graphs. Without domain-specific graph preprocessing, GTNs achieved the
best performance in all three benchmark node classification tasks against the state-
of-the-art methods that require pre-defined meta-paths from domain knowledge.

## 1 Introduction

In recent years, Graph Neural Networks (GNNs) have been widely adopted in various tasks over
graphs, such as graph classification [11, 21, 40], link prediction [18, 30, 42] and node classification
[3, 14, 33]. The representation learnt by GNNs has been proven to be effective in achieving state-of-
the-art performance in a variety of graph datasets such as social networks [7, 14, 35], citation networks
[19, 33], functional structure of brains [20], recommender systems [1, 27, 39]. The underlying graph
structure is utilized by GNNs to operate convolution directly on graphs by passing node features
[12, 14] to neighbors, or perform convolution in the spectral domain using the Fourier basis of a
given graph, i.e., eigenfunctions of the Laplacian operator [9, 15, 19].

However, one limitation of most GNNs is that they assume the graph structure to operate GNNs on is
*fixed* and *homogeneous*. Since the graph convolutions discussed above are determined by the fixed
graph structure, a noisy graph with missing/spurious connections results in ineffective convolution
with wrong neighbors on the graph. In addition, in some applications, constructing a graph to operate
GNNs is not trivial. For example, a citation network has multiple types of nodes (e.g., authors,
papers, conferences) and edges defined by their relations (e.g., author-paper, paper-conference),
and it is called a *heterogeneous* graph. A naïve approach is to ignore the node/edge types and
treat them as in a *homogeneous* graph (a standard graph with one type of nodes and edges). This,
apparently, is suboptimal since models cannot exploit the type information. A more recent remedy is
to manually design meta-paths, which are paths connected with heterogeneous edges, and transform

---

a heterogeneous graph into a *homogeneous* graph defined by the meta-paths. Then conventional GNNs can operate on the transformed homogeneous graphs [37, 43]. This is a two-stage approach and requires hand-crafted meta-paths for each problem. The accuracy of downstream analysis can be significantly affected by the choice of these meta-paths.

Here, we develop Graph Transformer Network (GTN) that learns to transform a heterogeneous input graph into useful meta-path graphs for each task and learn node representation on the graphs in an end-to-end fashion. GTNs can be viewed as a graph analogue of Spatial Transformer Networks [16] which explicitly learn spatial transformations of input images or features. The main challenge to transform a heterogeneous graph into new graph structure defined by meta-paths is that meta-paths may have an arbitrary length and edge types. For example, author classification in citation networks may benefit from meta-paths which are Author-Paper-Author (APA) or Author-Paper-Conference-Paper-Author (APCPA). Also, the citation networks are directed graphs where relatively less graph neural networks can operate on. In order to address the challenges, we require a model that generates new graph structures based on composite relations connected with softly chosen edge types in a heterogeneous graph and learns node representations via convolution on the learnt graph structures for a given problem.

Our **contributions** are as follows: (i) We propose a novel framework Graph Transformer Networks, to learn a new graph structure which involves identifying useful meta-paths and multi-hop connections for learning effective node representation on graphs. (ii) The graph generation is interpretable and the model is able to provide insight on effective meta-paths for prediction. (iii) We prove the effectiveness of node representation learnt by Graph Transformer Networks resulting in the best performance against state-of-the-art methods that additionally use domain knowledge in all three benchmark node classification on heterogeneous graphs.

## 2   Related Works

**Graph Neural Networks.** In recent years, many classes of GNNs have been developed for a wide range of tasks. They are categorized into two approaches: spectral [5, 9, 15, 19, 22, 38] and non-spectral methods [7, 12, 14, 26, 29, 33]. Based on spectral graph theory, *Bruna et al.* [5] proposed a way to perform convolution in the spectral domain using the Fourier basis of a given graph. *Kipf et al.* [19] simplified GNNs using the first-order approximation of the spectral graph convolution. On the other hand, non-spectral approaches define convolution operations directly on the graph, utilizing spatially close neighbors. For instance, *Veličković et al.* [33] applies different weight matrices for nodes with different degrees and *Hamilton et al.* [14] has proposed learnable aggregator functions which summarize neighbors' information for graph representation learning.

**Node classification with GNNs.** Node classification has been studied for decades. Conventionally, hand-crafted features have been used such as simple graph statistics [2], graph kernel [34], and engineered features from a local neighbor structure [23]. These features are not flexible and suffer from poor performance. To overcome the drawback, recently node representation learning methods via random walks on graphs have been proposed in DeepWalk [28], LINE [32], and node2vec [13] with tricks from deep learning models (e.g., skip-gram) and have gained some improvement in performance. However, all of these methods learn node representation solely based on the graph structure. The representations are not optimized for a specific task. As CNNs have achieved remarkable success in representation learning, GNNs learn a powerful representation for given tasks and data. To improve performance or scalability, generalized convolution based on spectral convolution [4, 26], attention mechanism on neighbors [25, 33], subsampling [6, 7] and inductive representation for a large graph [14] have been studied. Although these methods show outstanding results, all these methods have a common limitation which only deals with a *homogeneous* graph.

However, many real-world problems often cannot be represented by a single homogeneous graph. The graphs come as a heterogeneous graph with various types of nodes and edges. Since most GNNs are designed for a single homogeneous graph, one simple solution is a two-stage approach. Using meta-paths that are the composite relations of multiple edge types, as a preprocessing, it converts the heterogeneous graph into a homogeneous graph and then learns representation. The metapath2vec [10] learns graph representations by using meta-path based random walk and HAN [37] learns graph representation learning by transforming a heterogeneous graph into a homogeneous graph constructed by meta-paths. However, these approaches manually select meta-paths by domain experts and thus might not be able to capture all meaningful relations for each problem. Also, performance can be

significantly affected by the choice of meta-paths. Unlike these approaches, our Graph Transformer Networks can operate on a heterogeneous graph and transform the graph for tasks while learning node representation on the transformed graphs in an end-to-end fashion.

# 3 Method

The goal of our framework, Graph Transformer Networks, is to generate new graph structures and learn node representations on the learned graphs simultaneously. Unlike most CNNs on graphs that assume the graph is given, GTNs seek for new graph structures using multiple candidate adjacency matrices to perform more effective graph convolutions and learn more powerful node representations. Learning new graph structures involves identifying useful meta-paths, which are paths connected with heterogeneous edges, and multi-hop connections. Before introducing our framework, we briefly summarize the basic concepts of meta-paths and graph convolution in GCNs.

## 3.1 Preliminaries

One input to our framework is multiple graph structures with different types of nodes and edges. Let $\mathcal{T}^v$ and $\mathcal{T}^e$ be the set of node types and edge types respectively. The input graphs can be viewed as a heterogeneous graph [31] $G = (V, E)$, where $V$ is a set of nodes, $E$ is a set of observed edges with a node type mapping function $f_v : V \to \mathcal{T}^v$ and an edge type mapping function $f_e : E \to \mathcal{T}^e$. Each node $v_i \in V$ has one node type, i.e., $f_v(v_i) \in \mathcal{T}^v$. Similarly, for $e_{ij} \in E$, $f_e(e_{ij}) \in \mathcal{T}^e$. When $|\mathcal{T}^e| = 1$ and $|\mathcal{T}^v| = 1$, it becomes a standard graph. In this paper, we consider the case of $|\mathcal{T}^e| > 1$. Let $N$ denotes the number of nodes, i.e., $|V|$. The heterogeneous graph can be represented by a set of adjacency matrices $\{A_k\}_{k=1}^K$ where $K = |\mathcal{T}^e|$, and $A_k \in \mathbf{R}^{N \times N}$ is an adjacency matrix where $A_k[i, j]$ is non-zero when there is a k-th type edge from $j$ to $i$. More compactly, it can be written as a tensor $\mathbb{A} \in \mathbf{R}^{N \times N \times K}$. We also have a feature matrix $X \in \mathbf{R}^{N \times D}$ meaning that the D-dimensional input feature given for each node.

**Meta-Path [37]** denoted by $p$ is a path on the heterogeneous graph $G$ that is connected with heterogeneous edges, i.e., $v_1 \xrightarrow{t_1} v_2 \xrightarrow{t_2} \dots \xrightarrow{t_l} v_{l+1}$, where $t_l \in \mathcal{T}^e$ denotes an $l$-th edge type of meta-path. It defines a composite relation $R = t_1 \circ t_2 \dots \circ t_l$ between node $v_1$ and $v_{l+1}$, where $R_1 \circ R_2$ denotes the composition of relation $R_1$ and $R_2$. Given the composite relation $R$ or the sequence of edge types $(t_1, t_2, \dots, t_l)$, the adjacency matrix $A_{\mathcal{P}}$ of the meta-path $P$ is obtained by the multiplications of adjacency matrices as

$$A_{\mathcal{P}} = A_{t_l} \dots A_{t_2} A_{t_1}. \tag{1}$$

The notion of meta-path subsumes multi-hop connections and new graph structures in our framework are represented by adjacency matrices. For example, the meta-path Author-Paper-Conference (APC), which can be represented as $A \xrightarrow{AP} P \xrightarrow{PC} C$, generates an adjacency matrix $A_{APC}$ by the multipication of $A_{AP}$ and $A_{PC}$.

**Graph Convolutional network (GCN).** In this work, a graph convolutional network (GCN) [19] is used to learn useful representations for node classification in an end-to-end fashion. Let $H^{(l)}$ be the feature representations of the $l$th layer in GCNs, the forward propagation becomes

$$H^{(l+1)} = \sigma\left(\tilde{D}^{-\frac{1}{2}} \tilde{A} \tilde{D}^{-\frac{1}{2}} H^{(l)} W^{(l)}\right), \tag{2}$$

where $\tilde{A} = A + I \in \mathbf{R}^{N \times N}$ is the adjacency matrix $A$ of the graph $G$ with added self-connections, $\tilde{D}$ is the degree matrix of $\tilde{A}$, i.e., $\tilde{D}_{ii} = \sum_i \tilde{A}_{ij}$, and $W^{(l)} \in \mathbf{R}^{d \times d}$ is a trainable weight matrix. One can easily observe that the convolution operation across the graph is determined by the given graph structure and it is not learnable except for the node-wise linear transform $H^{(l)} W^{(l)}$. So the convolution layer can be interpreted as the composition of a fixed convolution followed by an activation function $\sigma$ on the graph after a node-wise linear transformation. Since we learn graph structures, our framework benefits from the different convolutions, namely, $\tilde{D}^{-\frac{1}{2}} \tilde{A} \tilde{D}^{-\frac{1}{2}}$, obtained from learned multiple adjacency matrices. The architecture will be introduced later in this section. For a directed graph (i.e., asymmetric adjacency matrix), $\tilde{A}$ in (2) can be normalized by the inverse of in-degree diagonal matrix $D^{-1}$ as $H^{(l+1)} = \sigma(\tilde{D}^{-1} \tilde{A} H^{(l)} W^{(l)})$.

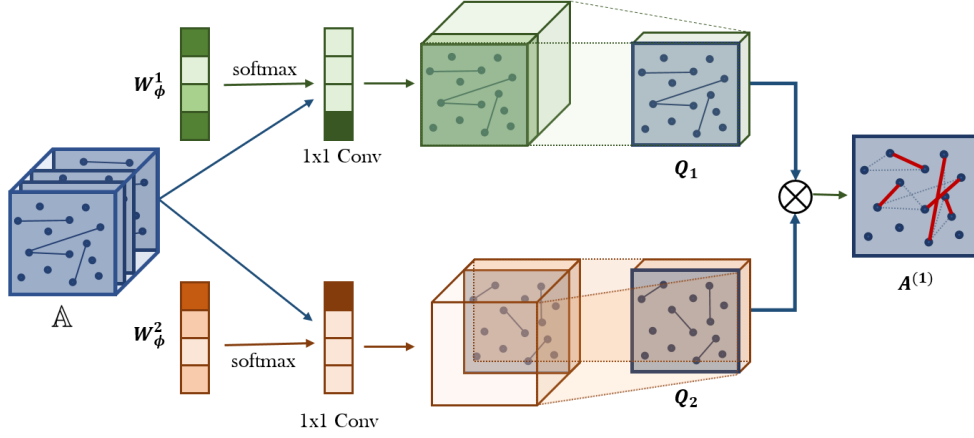

Figure 1: **Graph Transformer Layer** softly selects adjacency matrices (edge types) from the set of adjacency matrices $\mathbb{A}$ of a heterogeneous graph $G$ and learns a new meta-path graph represented by $A^{(1)}$ via the matrix multiplication of two selected adjacency matrices $Q_1$ and $Q_2$. The soft adjacency matrix selection is a weighted sum of candidate adjacency matrices obtained by $1 \times 1$ convolution with non-negative weights from softmax($W_\phi^1$).

## 3.2 Meta-Path Generation

Previous works [37, 43] require manually defined meta-paths and perform Graph Neural Networks on the meta-path graphs. Instead, our Graph Transformer Networks (GTNs) learn meta-paths for given data and tasks and operate graph convolution on the learned meta-path graphs. This gives a chance to find more useful meta-paths and lead to virtually various graph convolutions using multiple meta-path graphs.

The new meta-path graph generation in Graph Transformer (GT) Layer in Fig. 1 has two components. First, GT layer softly selects two graph structures $Q_1$ and $Q_2$ from candidate adjacency matrices $\mathbb{A}$. Second, it learns a new graph structure by the composition of two relations (i.e., matrix multiplication of two adjacency matrices, $Q_1 Q_2$).

It computes the convex combination of adjacency matrices as $\sum_{t_l \in \mathcal{T}^e} \alpha_{t_l}^{(l)} A_{t_l}$ in (4) by 1x1 convolution as in Fig. 1 with the weights from softmax function as

$$Q = F(\mathbb{A}; W_\phi) = \phi(\mathbb{A}; \text{softmax}(W_\phi)), \tag{3}$$

where $\phi$ is the convolution layer and $W_\phi \in \mathbf{R}^{1 \times 1 \times K}$ is the parameter of $\phi$. This trick is similar to channel attention pooling for low-cost image/action recognition in [8]. Given two softly chosen adjacency matrices $Q_1$ and $Q_2$, the meta-path adjacency matrix is computed by matrix multiplication, $Q_1 Q_2$. For numerical stability, the matrix is normalized by its degree matrix as $A^{(l)} = D^{-1} Q_1 Q_2$.

Now, we need to check whether GTN can learn an arbitrary meta-path with respect to edge types and path length. The adjacency matrix of arbitrary length $l$ meta-paths can be calculated by

$$A_P = \left( \sum_{t_1 \in \mathcal{T}^e} \alpha_{t_1}^{(1)} A_{t_1} \right) \left( \sum_{t_2 \in \mathcal{T}^e} \alpha_{t_2}^{(2)} A_{t_2} \right) \dots \left( \sum_{t_l \in \mathcal{T}^e} \alpha_{t_l}^{(l)} A_{t_l} \right) \tag{4}$$

where $A_P$ denotes the adjacency matrix of meta-paths, $\mathcal{T}^e$ denotes a set of edge types and $\alpha_{t_l}^{(l)}$ is the weight for edge type $t_l$ at the $l$th GT layer. When $\alpha$ is not one-hot vector, $A_P$ can be seen as the weighted sum of all length-$l$ meta-path adjacency matrices. So a stack of $l$ GT layers allows to learn arbitrary length $l$ meta-path graph structures as the architecture of GTN shown in Fig. 2. One issue with this construction is that adding GT layers always increase the length of meta-path and this does not allow the original edges. In some applications, both long meta-paths and short meta-paths are important. To learn short and long meta-paths including original edges, we include the identity

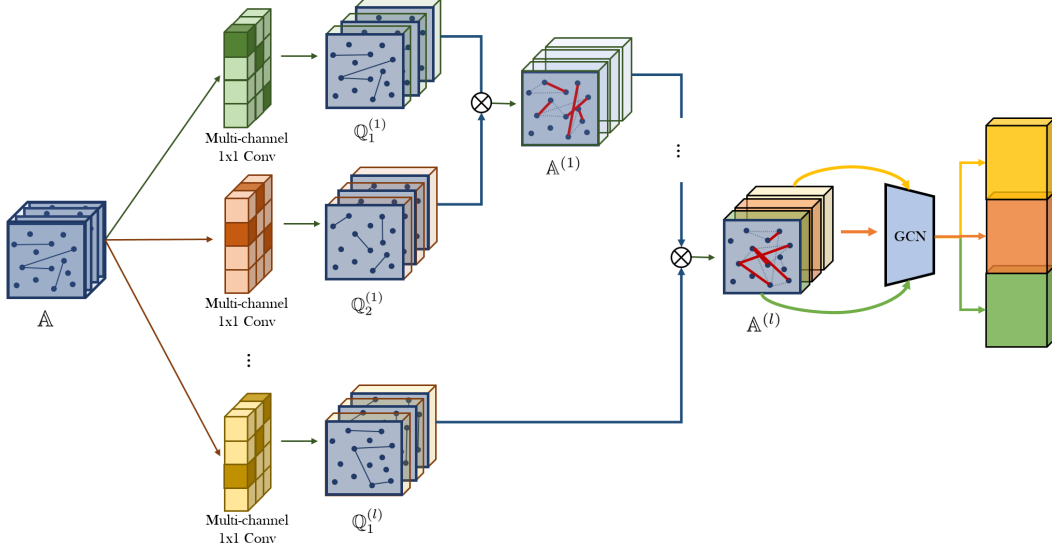

Figure 2: Graph Transformer Networks (GTNs) learn to generate a set of new meta-path adjacency matrices $\mathbb{A}^{(l)}$ using GT layers and perform graph convolution as in GCNs on the new graph structures. Multiple node representations from the same GCNs on multiple meta-path graphs are integrated by concatenation and improve the performance of node classification. $\mathbb{Q}_1^{(l)}$ and $\mathbb{Q}_2^{(l)} \in \mathbf{R}^{N \times N \times C}$ are intermediate adjacency tensors to compute meta-paths at the $l$th layer.

matrix $I$ in $\mathbb{A}$, i.e., $A_0 = I$. This trick allows GTNs to learn any length of meta-paths up to $l + 1$ when $l$ GT layers are stacked.

## 3.3 Graph Transformer Networks

We here introduce the architecture of Graph Transformer Networks. To consider multiple types of meta-paths simultaneously, the output channels of $1 \times 1$ convolution in Fig. 1 is set to $C$. Then, the GT layer yields a set of meta-paths and the intermediate adjacency matrices $Q_1$ and $Q_2$ become adjacency tensors $\mathbb{Q}_1$ and $\mathbb{Q}_2 \in \mathbf{R}^{N \times N \times C}$ as in Fig.2. It is beneficial to learn different node representations via multiple different graph structures. After the stack of $l$ GT layers, a GCN is applied to each channel of meta-path tensor $\mathbb{A}^{(l)} \in \mathbf{R}^{N \times N \times C}$ and multiple node representations are concatenated as

$$Z = \mathop{\Big\|}_{i=1}^{C} \sigma(\tilde{D}_i^{-1} \tilde{A}_i^{(l)} X W), \tag{5}$$

where $\big\|$ is the concatenation operator, C denotes the number of channels, $\tilde{A}_i^{(l)} = A_i^{(l)} + I$ is the adjacency matrix from the $i$th channel of $\mathbb{A}^{(l)}$, $\tilde{D}_i$ is the degree matrix of $\tilde{A}_i^{(l)}$, $W \in \mathbf{R}^{d \times d}$ is a trainable weight matrix shared across channels and $X \in \mathbf{R}^{N \times d}$ is a feature matrix. $Z$ contains the node representations from $C$ different meta-path graphs with variable, at most $l + 1$, lengths. It is used for node classification on top and two dense layers followed by a softmax layer are used. Our loss function is a standard cross-entropy on the nodes that have ground truth labels. This architecture can be viewed as an ensemble of GCNs on multiple meta-path graphs learnt by GT layers.

## 4 Experiments

In this section, we evaluate the benefits of our method against a variety of state-of-the-art models on node classification. We conduct experiments and analysis to answer the following research questions: **Q1.** Are the new graph structures generated by GTN effective for learning node representation? **Q2.** Can GTN adaptively produce a variable length of meta-paths depending on datasets? **Q3.** How can we interpret the importance of each meta-path from the adjacency matrix generated by GTNs?

Table 1: Datasets for node classification on heterogeneous graphs.

| Dataset | # Nodes | # Edges | # Edge type | # Features | # Training | # Validation | # Test |
|---------|---------|---------|-------------|------------|------------|--------------|--------|
| DBLP | 18405 | 67946 | 4 | 334 | 800 | 400 | 2857 |
| ACM | 8994 | 25922 | 4 | 1902 | 600 | 300 | 2125 |
| IMDB | 12772 | 37288 | 4 | 1256 | 300 | 300 | 2339 |

**Datasets.** To evaluate the effectiveness of meta-paths generated by Graph Transformer Networks, we used heterogeneous graph datasets that have multiple types of nodes and edges. The main task is node classification. We use two citation network datasets DBLP and ACM, and a movie dataset IMDB. The statistics of the heterogeneous graphs used in our experiments are shown in Table 1. DBLP contains three types of nodes (papers (P), authors (A), conferences (C)), four types of edges (PA, AP, PC, CP), and research areas of authors as labels. ACM contains three types of nodes (papers (P), authors (A), subject (S)), four types of edges (PA, AP, PS, SP), and categories of papers as labels. Each node in the two datasets is represented as bag-of-words of keywords. On the other hand, IMDB contains three types of nodes (movies (M), actors (A), and directors (D)) and labels are genres of movies. Node features are given as bag-of-words representations of plots.

**Implementation details.** We set the embedding dimension to 64 for all the above methods for a fair comparison. The Adam optimizer was used and the hyperparameters (e.g., learning rate, weight decay etc.) are respectively chosen so that each baseline yields its best performance. For random walk based models, a walk length is set to 100 per node for 1000 iterations and the window size is set to 5 with 7 negative samples. For GCN, GAT, and HAN, the parameters are optimized using the validation set, respectively. For our model GTN, we used three GT layers for DBLP and IMDB datasets, two GT layers for ACM dataset. We initialized parameters for $1 \times 1$ convolution layers in the GT layer with a constant value. Our code is publicly available at `https://github.com/seongjunyun/Graph_Transformer_Networks`.

## 4.1 Baselines

To evaluate the effectiveness of representations learnt by the Graph Transformer Networks in node classification, we compare GTNs with conventional random walk based baselines as well as state-of-the-art GNN based methods.

**Conventional Network Embedding methods** have been studied and recently DeepWalk [28] and metapath2vec [10] have shown predominant performance among random walk based approaches. DeepWalk is a random walk based network embedding method which is originally designed for homogeneous graphs. Here we ignore the heterogeneity of nodes/edges and perform DeepWalk on the whole heterogeneous graph. However, metapath2vec is a heterogeneous graph embedding method which performs meta-path based random walk and utilizes skip-gram with negative sampling to generate embeddings.

**GNN-based methods** We used the GCN [19], GAT [33], and HAN [37] as GNN based methods. GCN is a graph convolutional network which utilizes a localized first-order approximation of the spectral graph convolution designed for the symmetric graphs. Since our datasets are directed graphs, we modified degree normalization for asymmetric adjacency matrices, i.e., $\tilde{D}^{-1}\tilde{A}$ rather than $\tilde{D}^{-1/2}\tilde{A}\tilde{D}^{-1/2}$. GAT is a graph neural network which uses the attention mechanism on the homogeneous graphs. We ignore the heterogeneity of node/edges and perform GCN and GAT on the whole graph. HAN is a graph neural network which exploits manually selected meta-paths. This approach requires a manual transformation of the original graph into sub-graphs by connecting vertices with pre-defined meta-paths. Here, we test HAN on the selected sub-graphs whose nodes are linked with meta-paths as described in [37].

## 4.2 Results on Node Classification

**Effectiveness of the representation learnt from new graph structures.** Table 2. shows the performances of GTN and other node classification baselines. By analysing the result of our experiment, we will answer the research **Q1** and **Q2**. We observe that our GTN achieves the highest performance on all the datasets against all network embedding methods and graph neural network methods.

Table 2: Evaluation results on the node classification task (F1 score).

|  | DeepWalk | metapath2vec | GCN | GAT | HAN | $GTN_{-I}$ | **GTN (proposed)** |
|---|---|---|---|---|---|---|---|
| DBLP | 63.18 | 85.53 | 87.30 | 93.71 | 92.83 | 93.91 | **94.18** |
| ACM | 67.42 | 87.61 | 91.60 | 92.33 | 90.96 | 91.13 | **92.68** |
| IMDB | 32.08 | 35.21 | 56.89 | 58.14 | 56.77 | 52.33 | **60.92** |

GNN-based methods, *e.g.*, GCN, GAT, HAN, and the GTN perform better than random walk-based network embedding methods. Furthermore, the GAT usually performs better than the GCN. This is because the GAT can specify different weights to neighbor nodes while the GCN simply averages over neighbor nodes. Interestingly, though the HAN is a modified GAT for a heterogeneous graph, the GAT usually performs better than the HAN. This result shows that using the pre-defined meta-paths as the HAN may cause adverse effects on performance. In contrast, Our GTN model achieved the best performance compared to all other baselines on all the datasets even though the GTN model uses only one GCN layer whereas GCN, GAT and HAN use at least two layers. It demonstrates that the GTN can learn a new graph structure which consists of useful meta-paths for learning more effective node representation. Also compared to a simple meta-path adjacency matrix with a constant in the baselines, *e.g.*, HAN, the GTN is capable of assigning variable weights to edges.

**Identify matrix in $\mathbb{A}$ to learn variable-length meta-paths**. As mentioned in Section 3.2, the identity matrix is included in the candidate adjacency matrices $\mathbb{A}$. To verify the effect of identity matrix, we trained and evaluated another model named $GTN_{-I}$ as an ablation study. the $GTN_{-I}$ has exactly the same model structure as GTN but its candidate adjacency matrix $\mathbb{A}$ doesn't include an identity matrix. In general, the $GTN_{-I}$ consistently performs worse than the GTN. It is worth to note that the difference is greater in IMDB than DBLP. One explanation is that the length of meta-paths $GTN_{-I}$ produced is not effective in IMDB. As we stacked 3 layers of GTL, $GTN_{-I}$ always produce 4-length meta-paths. However shorter meta-paths (e.g. MDM) are preferable in IMDB.

### 4.3 Interpretation of Graph Transformer Networks

We examine the transformation learnt by GTNs to discuss the question interpretability **Q3**. We first describe how to calculate the importance of each meta-path from our GT layers. For the simplicity, we assume the number of output channels is one. To avoid notational clutter, we define a shorthand notation $\alpha \cdot \mathbb{A} = \sum_k^K \alpha_k A_k$ for a convex combination of input adjacency matrices. The $l$th GT layer in Fig. 2 generates an adjacency matrix $A^{(l)}$ for a new meta-path graph using the previous layer's output $A^{(l-1)}$ and input adjacency matrices $\alpha^{(l)} \cdot \mathbb{A}$ as follows:

$$A^{(l)} = \left(D^{(l-1)}\right)^{-1} A^{(l-1)} \left(\sum_i^K \alpha_i^{(l)} A_i\right), \tag{6}$$

where $D^{(l)}$ denotes a degree matrix of $A^{(l)}$, $A_i$ denotes the input adjacency matrix for an edge type $i$ and $\alpha_i$ denotes the weight of $A_i$. Since we have two convex combinations at the first layer as in Fig. 1, we denote $\alpha^{(0)} = \text{softmax}(W_\phi^1)$, $\alpha^{(1)} = \text{softmax}(W_\phi^2)$. In our GTN, the meta-path tensor from the previous tensor is reused for $\mathbb{Q}_1^l$, we only need $\alpha^{(l)} = \text{softmax}(W_\phi^2)$ for each layer to calculate $\mathbb{Q}_2^l$. Then, the new adjacency matrix from the $l$th GT layer can be written as

$$A^{(l)} = \left(D^{(l-1)}\right)^{-1} \ldots \left(D^{(1)}\right)^{-1} \left((\alpha^{(0)} \cdot \mathbb{A})(\alpha^{(1)} \cdot \mathbb{A})(\alpha^{(2)} \cdot \mathbb{A}) \ldots (\alpha^{(l)} \cdot \mathbb{A})\right) \tag{7}$$

$$= \left(D^{(l-1)}\right)^{-1} \ldots \left(D^{(1)}\right)^{-1} \left(\sum_{t_0, t_1, \ldots, t_l \in \mathcal{T}^e} \alpha_{t_0}^{(0)} \alpha_{t_1}^{(1)} \ldots \alpha_{t_l}^{(l)} A_{t_0} A_{t_1} \ldots A_{t_l}\right), \tag{8}$$

where $\mathcal{T}^e$ denotes a set of edge types and $\alpha_{t_l}^{(l)}$ is an attention score for edge type $t_l$ at the $l$th GT layer. So, $A^{(l)}$ can be viewed as a weighted sum of all meta-paths including 1-length (original edges) to $l$-length meta-paths. The contribution of a meta-path $t_l, t_{l-1}, \ldots, t_0$, is obtained by $\prod_{i=0}^l \alpha_{t_i}^{(i)}$.

Table 3: Comparison with predefined meta-paths and top-ranked meta-paths by GTNs. Our model found important meta-paths that are consistent with pre-defined meta-paths between target nodes (a type of nodes with labels for node classifications). Also, new relevant meta-paths between all types of nodes are discovered by GTNs.

| Dataset | Predefined Meta-path | Meta-path learnt by GTNs | |
|---|---|---|---|
| | | Top 3 (between target nodes) | Top 3 (all) |
| DBLP | APCPA, APA | APCPA, APAPA, APA | CPCPA, APCPA, CP |
| ACM | PAP, PSP | PAP, PSP | APAP, APA, SPAP |
| IMDB | MAM, MDM | MDM, MAM, MDMDM | DM, AM, MDM |

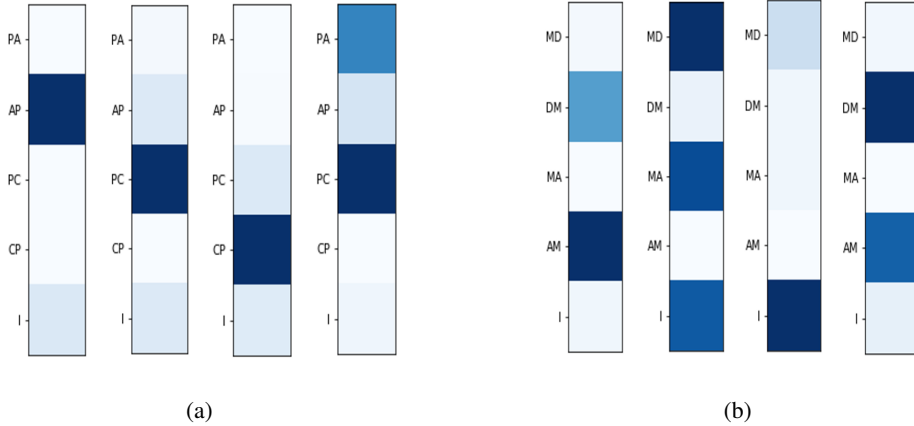

(a)  (b)

Figure 3: After applying softmax function on 1x1 conv filter $W_\phi^i$ (i: index of layer) in Figure 1, we visualized this attention score of adjacency matrix (edge type) in DBLP (left) and IMDB (right) datasets. (a) Respectively, each edge indicates (Paper-Author), (Author-Paper), (Paper-Conference), (Conference-Paper), and identity matrix. (b) Edges in IMDB dataset indicates (Movie-Director), (Director-Movie), (Movie-Actor), (Actor-Movie), and identity matrix.

Now we can interpret new graph structures learnt by GTNs. The weight $\prod_{i=0}^{l} \alpha_{t_i}^{(i)}$ for a meta-path $(t_0, t_1, \ldots t_l)$ is an attention score and it provides the importance of the meta-path in the prediction task. In Table 3 we summarized predefined meta-paths, that are widely used in literature, and the meta-paths with high attention scores learnt by GTNs.

As shown in Table 3, between target nodes, that have class labels to predict, the predefined meta-paths by domain knowledge are consistently top-ranked by GTNs as well. This shows that GTNs are capable of learning the importance of meta-paths for tasks. More interestingly, GTNs discovered important meta-paths that are not in the predefined meta-path set. For example, in the DBLP dataset GTN ranks CPCPA as most importance meta-paths, which is not included in the predefined meta-path set. It makes sense that author's research area (label to predict) is relevant to the venues where the author publishes. We believe that the interpretability of GTNs provides useful insight in node classification by the attention scores on meta-paths.

Fig.3 shows the attention scores of adjacency matrices (edge type) from each Graph Transformer Layer. Compared to the result of DBLP, identity matrices have higher attention scores in IMDB. As discussed in Section 3.3, a GTN is capable of learning shorter meta-paths than the number of GT layers, which they are more effective as in IMDB. By assigning higher attention scores to the identity matrix, the GTN tries to stick to the shorter meta-paths even in the deeper layer. This result demonstrates that the GTN has ability to adaptively learns most effective meta-path length depending on the dataset.

# 5   Conclusion

We proposed Graph Transformer Networks for learning node representation on a heterogeneous graph. Our approach transforms a heterogeneous graph into multiple new graphs defined by meta-paths with arbitrary edge types and arbitrary length up to one less than the number of Graph Transformer layers while it learns node representation via convolution on the learnt meta-path graphs. The learnt graph structures lead to more effective node representation resulting in state-of-the art performance, without any predefined meta-paths from domain knowledge, on all three benchmark node classification on heterogeneous graphs. Since our Graph Transformer layers can be combined with existing GNNs, we believe that our framework opens up new ways for GNNs to optimize graph structures by themselves to operate convolution depending on data and tasks without any manual efforts. Interesting future directions include studying the efficacy of GT layers combined with different classes of GNNs rather than GCNs. Also, as several heterogeneous graph datasets have been recently studied for other network analysis tasks, such as link prediction [36, 41] and graph classification [17, 24], applying our GTNs to the other tasks can be interesting future directions.

# 6   Acknowledgement

This work was supported by the National Research Foundation of Korea (NRF) grant funded by the Korea government (MSIT) (NRF-2019R1G1A1100626, NRF-2016M3A9A7916996, NRF-2017R1A2A1A17069645).

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
