[Supplementary Material 1]



# Graph Transformer Networks (NeurIPS 2019)

Seongjun Yun, Minbyul Jeong, Raehyun Kim, Jaewoo Kang, Hyunwoo J. Kim

## Spatial Transformer Networks

(a)   (b)   (c)   (d)

A spatial transformer that crops out and scale-normalizes the appropriate region can simplify the subsequent classification task, and lead to superior classification performance

Spatial Transformer Networks, NIPS 2015, M Jaderberg et al.

# Heterogeneous Graph

Heterogeneous graph is a graph which contains different types of nodes and edges

# Meta-path

A meta-path is a path consisting of a sequence of relations defined between different object types

| Datasets | Examples of meta-path |
|---|---|
| BlogCatalog | Blog $\xrightarrow{has}$ Tag $\xrightarrow{has^{-1}}$ Blog |
| | Blog $\xrightarrow{written\_by}$ User $\xrightarrow{written\_by^{-1}}$ Blog |
| | Blog $\xrightarrow{written\_by}$ User $\xrightarrow{friend}$ User $\xrightarrow{written\_by^{-1}}$ Blog |
| DBLP | Paper $\xrightarrow{has}$ Term $\xrightarrow{has^{-1}}$ Paper |
| | Paper $\xrightarrow{written\_by}$ Author $\xrightarrow{written\_by^{-1}}$ Paper |
| | Paper $\xrightarrow{written\_by}$ Author $\xrightarrow{written\_by^{-1}}$ Paper $\xrightarrow{has}$ Term $\xrightarrow{has^{-1}}$ Paper |
| Chemical Compound | Compound $\xrightarrow{bind}$ Gene $\xrightarrow{PPI}$ Gene $\xrightarrow{bind^{-1}}$ Compound |
| | Compound $\xrightarrow{treat}$ Disease $\xrightarrow{cause^{-1}}$ Gene $\xrightarrow{bind^{-1}}$ Compound |
| | Compound $\xrightarrow{bind}$ Gene $\xrightarrow{has}$ Pathway $\xrightarrow{has^{-1}}$ Gene $\xrightarrow{bind^{-1}}$ Compound |

Actor

Movie

Director

Movie-Actor-Moive

Movie-Director-Moive

(c) Meta-path

(a) Node Type (b) Heterogeneous Graph (d) Meta-path based Neighbors

Heterogeneous Graph Attention Network, WWW 2019, Xiao Wang et al.

# Transform a Graph into new Graphs using Meta-Paths

● : Target node

*Meta path*

Previous works about graph neural networks leveraged useful meta-path which selected **manually by domain experts**.

# Transform a Graph into new Graphs using Meta-Paths

● : Target node

Meta path

Previous works about graph neural networks leveraged useful meta-path which selected **manually by domain experts**.

Can model **learn to transform** an original graph into a new graph which involves only useful connections for task?

# Multiplication of Adjacency Matrices for Generating Meta-Paths

Edge Type: A,B

Graph

A      B              A - B

Adjacency

Matrix

$A_A$    X    $A_B$          $A_{AB}$

# Graph Transformer Layer

Graph Transformer Layer (GTL) softly **selects adjacency matrices** (edge types) from the set of adjacency matrices and **generate** a new meta-path graph via the **matrix multiplication** of two selected adjacency matrices.

# Graph Transformer Networks

Graph Transformer Networks (GTNs) learn to **generate a set of new meta-path adjacency matrices** using GT layers and perform graph convolution as in GCNs on the new graph structures.

Q1) Are the new graph structures generated by GTN effective for learning node representation?

| Dataset | # Nodes | # Edges | # Edge type | # Features | # Training | # Validation | # Test |
|---------|---------|---------|-------------|------------|------------|--------------|--------|
| DBLP | 18405 | 67946 | 4 | 334 | 800 | 400 | 2857 |
| ACM | 8994 | 25922 | 4 | 1902 | 600 | 300 | 2125 |
| IMDB | 12624 | 37288 | 4 | 1256 | 300 | 300 | 2339 |

| | DeepWalk | metapath2vec | GCN | GAT | HAN | GTN$_{-I}$ | GTN (proposed) |
|------|----------|--------------|-------|-------|-------|---------|----------------|
| DBLP | 63.18 | 85.53 | 87.30 | 93.71 | 92.83 | 93.91 | **94.18** |
| ACM | 67.42 | 87.61 | 91.60 | 92.33 | 90.96 | 91.13 | **92.68** |
| IMDB | 32.08 | 35.21 | 56.89 | 58.14 | 56.77 | 52.33 | **60.92** |

- Graph Transformer Networks (GTNs) achieves the **highest performance** on all the datasets against all network embedding methods and graph neural network methods
- GTNs performs **better** than HAN which uses the **pre-defined meta paths**.

Q2) Can GTN adaptively produce a variable length of meta-paths depending on datasets?

The attention score of adjacency matrix (edge type) from each Graph Transformer Layer

(a)
DBLP

(b)
IMDB

- By assigning **higher** attention scores to **the identity matrix (I)**, GTN tries to stick to the shorter meta-paths even in the deeper layer.
- GTN has ability to **adaptively learn most effective meta-path length** depending on the dataset

Q3) How can we interpret the importance of each meta-path from the adjacency matrix generated by GTNs?

Comparison with predefined paths and top-ranked meta-paths by GTNs

| Dataset | Predefined Meta-path | Meta-path learnt by GTNs | |
|---------|---------|---------|---------|
| | | Top 3 (between target nodes) | Top 3 (all) |
| DBLP | APCPA, APA | APCPA, APAPA, APA | APCPC, APCPA, PC |
| ACM | PAP, PSP | PAP, PSP | PAPA, APA, PAPS |
| IMDB | MAM, MDM | MDM, MAM, MDMDM | DM, AM, MDM |

- The predefined meta-paths by domain knowledge are consistently top-ranked by GTNs as well.
- Our GTNs are capable to learn the importance of meta-paths for tasks

## Scalability

1. Sample neighborhood

2. Aggregate feature information from neighbors

3. Predict graph context and label using aggregated information

Inductive representation learning on large graphs, NIPS 2017, Hamilton Will et al.

[Supplementary Material 2]

# Graph Transformer Networks

Seoungjun Yun, Minbyul Jeong, Raehyun Kim, Jaewoo Kang*, Hyunwoo J. Kim*
Department of Computer Science and Engineering, Korea University, Seoul, Korea
{ ysj5419, minbyuljeong, raehyun, kangj, hyunwoojkim } @ korea.ac.kr

## Background & Motivations

**Node type**

Author

Paper

Conference

**Heterogeneous Graph**

**Meta-Path**

Heterogeneous Graph → EXPERT ADVICE → Meta-Path

## Contributions

1. We propose a novel framework **Graph Transformer Networks**, to **learn a new graph structure** which involves identifying useful meta-paths and multi-hop connections for learning effective node representation on graphs.

2. The graph generation is **interpretable** and the model is able to provide insight on **effective meta-paths** for prediction.

3. We prove the effectiveness of node representation learnt by Graph Transformer Networks resulting in **the best performance** against state-of-the-art methods that additionally use domain knowledge in all three benchmark **node classification on heterogeneous graphs**.

## Our Approach

*Input Graph*    ***Graph Transformer** layer*    *Output Graph*

$$Q = F(\mathbb{A}; W_\phi) = \phi(\mathbb{A}; \mathrm{softmax}(W_\phi))$$
$$A^{(l)} = D^{-1} Q_1 Q_2$$

Edge type $A$ → Adjacency Matrix $Q_1$

Edge type $B$ → Adjacency Matrix $Q_2$

Edge type A-$B$ → Multiplication of two Matrices $Q_1 Q_2$

***Graph Transformer** Networks*

Multi-channel 1x1 Conv $\mathbb{Q}_1^{(1)}$

Multi-channel 1x1 Conv $\mathbb{Q}_2^{(1)}$

Multi-channel 1x1 Conv $\mathbb{Q}_1^{(l)}$

$$Z = \Big\|_{i=1}^{C} \sigma(\tilde{D}_i^{-1} \tilde{A}_i^{(l)} X W)$$

## Analysis

Q1) Are the new graph structures generated by GTN effective for learning node representation?

| | DeepWalk | metapath2vec | GCN | GAT | HAN | GTN$_{-I}$ | GTN (proposed) |
|---|---|---|---|---|---|---|---|
| DBLP | 63.18 | 85.53 | 87.30 | 93.71 | 92.83 | 93.91 | **94.18** |
| ACM | 67.42 | 87.61 | 91.60 | 92.33 | 90.96 | 91.13 | **92.68** |
| IMDB | 32.08 | 35.21 | 56.89 | 58.14 | 56.77 | 52.33 | **60.92** |

Q2) Can GTN adaptively produce a variable length of meta-paths depending on datasets?

(a)          (b)

Q3) Can we interpret the importance of each meta-path from the adjacency matrix generated by GTNs?

| Dataset | Predefined Meta-path | Meta-path learnt by GTNs | |
|---|---|---|---|
| | | Top 3 (between target nodes) | Top 3 (all) |
| DBLP | APCPA, APA | APCPA, APAPA, APA | APCPC, APCPA, PC |
| ACM | PAP, PSP | PAP, PSP | PAPA, APA, PAPS |
| IMDB | MAM, MDM | MDM, MAM, MDMDM | DM, AM, MDM |