[Reviews · NeurIPS 2019]

Reviewer 1



The paper's writing is clear and good. The interpretation of Graph Transformer Network (section 4.3) provides useful insights of the model design. The experiments look a bit lacking as it only demonstrates on two citation datasets and one movie datasets. It would be great to provide results on different datasets in different domains.

Reviewer 2



The paper revolves around the construction of appropriate models for heterogeneous graphs, where nodes and arcs are of different types. Overall, this is a fundamental topics for the massive development of Graph Neural Networks. The paper shows how to generate new graph structures and learn node representations by looking for new graph structures. The proposed Graph Transformer Network (GTN) learns to transform a heterogeneous input graph into useful meta-path graphs for each task. Overall, the paper provides remarkable results and faces quite a common problem in most interesting applications. This reviewer very much liked the idea behind meta-paths and the central idea behind the "attention scores". The presentation is acceptable but the curious reader can hardly grasp a clear foundation on the presentation of the model. Here are some questions that one very much would like to be addressed: 1. Pag 3, line 132: The case of directed graphs is simply supposed to be processed by changing eq. 2 so as there is no right-multiplication of \tilde{D}^{-1/2}. As far as I understand, in any case, this gives rise to the same computation as in non-directed graphs. However, one might consider that in case of directed graphs there are data flow computational schemes that make it possible to determine a state-based representation of the graph. Basically, the same feedforward computation that takes place in the neural map can be carried out at data level in directed graphs. 2. While the learning of the attention scores is a distinctive feature of the paper, at the same time, one might be worried about the nature of their convex combination. It requires the attention scores to sum up to one. When looking at eq. 8 (pag. 7) one might be worried about the \alpha product, since each term is remarkably lower than one! Isn't there a critical gradient vanishing? 3. Pag. 4 line 157 and pag. 7, line 235 refer to the idea of "choosing" A_0= I (identity matrix). It is claimed that this yields better results. The explanation at pag. 7 is not fully satisfactory. Basically, the identity matrix has an obvious meaning in terms of graph structure. However, what I'm missing is the reason why it should be included in the generation of the meta-paths. Couldn't be this connected to my previous comment (2)? 4. Eq. (5) is supposed to express by Z the mode representation emerging from different meta-paths. Why is the D matrix only left-multiplying the adjacency matrix? Maybe I'm missing an important technical detail since also my previous question 1 on directed graphs seems to be on on a related issue. 5. The authors might also better discuss and make references to attempts to deal with heterogeneous graphs characterized by different types of nodes/arcs. Basically, beginning with early studies on GNN, it has been realized that the stationarity of GNN can be overcome by involving different sets of parameters that are associated with different node/edge data type.

Reviewer 3



Originality: the proposed idea is inspired from the spatial transformer networks. Clarity: the notations of this paper are a bit confusing, especially for Sec 3.1 when describing heterogeneous graph and meta-paths. Significance: the proposed method provides a good way to interpret the generation of meta-paths and helps to understand the problem.

[Author Response · NeurIPS 2019]

We thank the reviewers for constructive comments and unanimous recommendation for acceptance. We address all the
concerns raised by the reviewers and clarify several points hoping for a more vigorous support for our paper.

**Q1[R2]. Normalization of Laplace operator for directed graphs by left-multiplication by** $D^{-1}$**.** This is a common
technique for directed graphs. On an undirected graph, the in-degree and out-degree of a given vertex are the same
and its adjacency matrix is symmetric. So $D^{-1/2}AD^{-1/2}$ performs both row-wise and column-wise normalization
of the adjacency matrix and yields a symmetric normalized Laplace operator as in (Kipf & Welling, 2016), i.e.,
$a'_{ij} = a_{ij}/\sqrt{d_i d_j}$, where $a'_{ij}$ is the $(i, j)$ element of the normalized $A$ by the degree of vertices $d_i$, $d_j$. However, for a
directed graph, the adjacency matrix is asymmetric and is usually normalized by in-degree of vertices. It is calculated
by the left-multiplication of the inverse of in-degree diagonal matrix $(D^{-1}A)$, i.e., $a'_{ij} = a_{ij}/d_i^{\text{in}}$ , where $d_i^{\text{in}} = \sum_j a_{ij}$
and $a_{ij}$ is the weight of the edge from vertex $j$ to vertex $i$ as defined in [1].

**Q2[R2]. Does the proposed method suffer from the vanishing gradient problem?** This is a great point. Actually,
we are aware that the gradient vanishing problem may occur in GTNs as any deep neural networks. However, in our
experiments, we have not observed any gradient vanishing problem. First, one main cause of the vanishing gradient
problem in standard CNNs is a sigmoid function. In our framework, GT Layers do not use sigmoid functions to
construct new meta-paths. Further, since too long meta-paths generally introduce more noise than signals in this paper,
less than or equal than 3 GT Layers are used as described in L.195- L.196.

**Q3[R2]. Why should the identity matrix** ($A_0 = I$) **be included for generating meta-paths?** As discussed in L.156-
L.159, the identity matrix in candidate adjacency matrices $\mathbb{A}$ allows to learn variable length meta-paths from length
1 to $l + 1$ when $l$ GT layers are stacked. Without the identity matrix in $\mathbb{A}$, GTN always generates meta-paths with a
fixed length of $l + 1$. This is suboptimal for some applications where both short and long meta-paths are important.
Further, the length of effective meta-paths varies across datasets. Including the identity matrix to learn variable length
meta-paths makes GTNs more robust to the choice of hyperparameters (e.g., the number of GT Layers) across datasets.

**Q4[R2]. More discussion about handling heterogeneous graphs.** As R2 mentioned, it is possible to handle hetero-
geneous graphs by associating a set of existing GNNs with different node/edge types. As a related work, we referred
R-GCN [28] in Introduction that models relational data. This line of works require manual association between GNNs
(or parameters) and node/edge types with domain knowledge. Including the recent work with a similar technique on
knowledge graphs [2], we will discuss more related works in the final version.

**Q5[R3, R1]. More comparison with recent baselines and more datasets.** To our knowledge, HAN[34] in our
baselines is the latest and greatest model to deal with heterogeneous graphs. It was published 10 days prior to our
submission at the WWW 2019. In our paper, we excluded methods that significantly underperform HAN and GAT
[31]. Heterogeneous graphs often occur in the wild but a few datasets have been studied in the literature for 'node
classification' that we mainly focused on in this work. We have demonstrated the superior performance of our GTN
against state-of-the-art methods on the three most widely used datasets in the paper. Instead of more experiments on
rarely used datasets in node classification, one possible direction is to evaluate GTNs for a different task. For example,
different heterogeneous graph datasets (e.g., Amazon-book, Last-FM, Yelp2018) have been recently studied for link
prediction as in [2]. We plan to conduct additional experiments on the datasets for link prediction. If accepted, we will
include additional experiments in the supplementary materials.

**Q6[R3]. Relation with Spatial Transformer Networks (STNs) and originality.** We discussed that STNs can be an
analogous model of GTNs among CNNs for images since both learn transformations of input data spaces. However,
new concepts on graphs, heterogeneous graphs (multiple input spaces) and meta-paths (composite relations), led to
substantial development. First, GTNs handle multiple graphs (input spaces) to learn meta-paths. This is quite different
from STNs which need to handle one input image space at a time. Second, GT layers softly select adjacency matrices
and perform matrix multiplications to yield new meta-path (composite relation) graphs. This technique unifies multi-hop
connections of homogeneous and heterogeneous graphs with variable length meta-paths. We believe that the remote
relationship between GTNs and STNs should not lead to the underestimation of the novelty of our work.

**Q7[R3]. Clearer notations and examples about heterogeneous graphs and meta-paths.** R1 liked that our writing
is 'clear and good'. We believe that given the complexity of heterogeneous graphs and meta-paths, our notations are
clear and consistent with other papers. As R3 pointed out, a more gentle introduction of meta-paths and heterogeneous
graphs with simple examples and illustrations helps readers. We agree that our examples provided in **Line 31.** and **Line**
**265.** may not be sufficient for some readers. In the final version, we will add some examples with the definition of
heterogeneous graphs and meta-paths in Section 3.1. Preliminaries.

**References**

[1] F. Bauer. Normalized graph Laplacians for directed graphs. *Linear Algebra and its Applications*, 436, 2012.

[2] X. Wang, X. He, Y. Cao, M. Liu, and T. Chua. KGAT: knowledge graph attention network for recommendation. *CoRR*.


[Meta-Review · NeurIPS 2019]

The paper proposes a graph transformer network for multirelational graphs. Thee network learns to transform the input graph into useful meta-path graphs for generating new graphs and for learning node representations. While the reviewers point out some weaknesses, we overall agree that this is an interesting piece of work addressing an important problem.